# Effects of Supplementation with the Standardized Extract of Saffron (affron^®^) on the Kynurenine Pathway and Melatonin Synthesis in Rats

**DOI:** 10.3390/antiox12081619

**Published:** 2023-08-16

**Authors:** Mario De la Fuente Muñoz, Marta Román-Carmena, Sara Amor, Ángel Luís García-Villalón, Alberto E. Espinel, Daniel González-Hedström, Miriam Granado García

**Affiliations:** 1Departamento de Fisiología, Facultad de Medicina, Universidad Autónoma de Madrid, 28029 Madrid, Spain; mario.delafuente@uam.es (M.D.l.F.M.); marta.romanc@uam.es (M.R.-C.); sara.amor@uam.es (S.A.); angeluis.villalon@uam.es (Á.L.G.-V.); 2Pharmactive Biotech Products S.L.U., Parque Científico de Madrid, Avenida del Doctor Severo Ochoa, 37 Local 4J, 28108 Alcobendas, Spain; aespinel@pharmactive.eu (A.E.E.); dgonzalez@pharmactive.eu (D.G.-H.); 3CIBER Fisiopatología de la Obesidad y Nutrición, Instituto de Salud Carlos III, 28029 Madrid, Spain

**Keywords:** saffron, sleep, melatonin, kynurenine, inflammation, antioxidant

## Abstract

Melatonin is a hormone that regulates sleep–wake cycles and is mainly synthesized in the pineal gland from tryptophan after its conversion into serotonin. Under normal conditions, less than 5% of tryptophan is reserved for the synthesis of serotonin and melatonin. The remaining 95% is metabolized in the liver through the kynurenine pathway. Increased levels of proinflammatory cytokines and cortisol increase the metabolism of tryptophan through the kynurenine pathway and reduce its availability for the synthesis of melatonin and serotonin, which may cause alterations in mood and sleep. The standardized saffron extract (affron^®^) has shown beneficial effects on mood and sleep disorders in humans, but the underlying mechanisms are not well understood. Thus, the aim of this work was to study the effects of affron^®^ supplementation on the kynurenine pathway and the synthesis of melatonin in rats. For this purpose, adult male Wistar rats were supplemented for 7 days with 150 mg/kg of affron^®^ or vehicle (2 mL/kg water) administered by gavage one hour before sleep. Affron^®^ supplementation reduced body weight gain and increased the circulating levels of melatonin, testosterone, and c-HDL. Moreover, animals supplemented with affron^®^ showed decreased serum levels of kynurenine, ET-1, and c-LDL. In the pineal gland, affron^®^ reduced *Il-6* expression and increased the expression of *Aanat*, the key enzyme for melatonin synthesis. In the liver, affron^®^ administration decreased the mRNA levels of the enzymes of the kynurenine pathway *Ido-2*, *Tod-2*, and *Aadat*, as well as the gene expression of *Il-1β* and *Tnf-α*. Finally, rats treated with affron^®^ showed increased mRNA levels of the antioxidant enzymes *Ho-1*, *Sod-1*, *Gsr*, and *Gpx-3*, both in the liver and in the pineal gland. In conclusion, affron^®^ supplementation reduces kynurenine levels and promotes melatonin synthesis in rats, possibly through its antioxidant and anti-inflammatory effects, making this extract a possible alternative for the treatment and/or prevention of mood and sleep disorders.

## 1. Introduction

Saffron is a spice obtained from the stigmas of *Crocus sativus* L., a plant of the Iridaceae family that is cultivated mainly in Asia and the Mediterranean area. Nowadays, the main saffron-producing countries are Iran, primarily, followed by Greece, Marocco, Spain, and India [1]. 

Since ancient times, saffron has been widely used for culinary purposes due to its organoleptic and aromatic properties. Regarding its chemical composition, it contains sugars (63%), proteins and amino acids (12%), lipids (5%), water (10%), and minerals (5%) and vitamins such as riboflavin (B2) and thiamine (B2) [1]. In addition, saffron also contains several bioactive compounds such as carotenoids, monoterpene aldehydes, monoterpenoids, isophorones, and flavonoids, which exert biological effects and provide numerous health benefits [2]. These compounds are present mainly in the stigmas, although other parts of the plant such as the tepals and the anthers are also reported to be a good source of bioactive substances [3,4]. Among them, the most important ones are the carotenoid crocins (also referred to as crocetin esters) and their bioavailable metabolite crocetin [5], which are responsible for the characteristic yellow color of saffron. Terpene safranal is responsible for saffron’s aroma, and the monoterpene glycoside picrocrocin provides the bitter taste to the spice and is the precursor of safranal [6,7]. In addition to their organoleptic characteristics, crocins, crocetin, and safranal are the main components responsible for the biological effects of saffron, including its anti-inflammatory and antioxidant properties [8,9]. Thanks to these properties, saffron has traditionally been widely used for several cosmetic and medicinal applications [10]. Currently, interest in saffron as a source of functional extracts has been confirmed by preclinical and clinical evidence [11,12]. 

One of the properties that have attained more attention in the last years is saffron’s ability to improve sleep quality and duration [13]. This is an important issue since sleep plays a critical role in human physiology, affecting not only brain activity but also other important physiological functions such as metabolism, immune, endocrine, and cardiovascular functions [14]. 

Sleep disorders affect approximately 10% to 20% of adults [15] and are associated with the development of both short- and long-term alterations. Among the short-term alterations are altered autonomic function and stress responsivity [16] and alterations in cognitive function that include impaired decision-making capacity, memory formation, executive function, and emotional reactivity [16,17]. The long-term consequences include an increased risk of suffering cardiovascular diseases, such as hypertension, atherosclerosis, myocardial ischemia [18], metabolic diseases, such as obesity and type II diabetes [19], cancer [20,21], and psychiatric disorders, such as anxiety or depression [16]. Indeed, sleep disturbance has been identified as a major cause of depressive and anxiety disorders [22]. Moreover, disrupted sleep is a frequent feature in patients suffering from depression or anxiety [23], making it difficult to discern between the cause and the consequence. 

Due to its high prevalence, in recent years, the use of drugs to treat insomnia and its associated co-morbidities has sharply increased. Among the most widely used drugs are antidepressants, anti-anxiolytics, anti-parkinsonians, anticonvulsants, anti-narcoleptics, anti-histamines, and hypnotics. These drugs are not exempt from side effects, including poor tolerance, daytime sleepiness, cognitive impairment, and the development of tolerance and dependency in the medium- and long-term, making it difficult to withdraw [24]. For this reason, a lot of effort is being invested in the search for new therapeutic agents, ideally of natural origin, that are effective for the treatment of sleep disorders and that show fewer side effects than conventional pharmacological treatments. Among these agents is melatonin, a hormone primarily produced by the pineal gland that regulates the sleep–wake cycles following a circadian rhythm. Melatonin is released into the bloodstream in the evening with the so-called dim light melatonin onset (DLMO). Its levels start to increase approximately two hours before the onset of natural sleep, and they peak approximately five hours later [25]. The use of both the melatonin and agonists of the melatonin M1 and M2 receptors to treat insomnia is sustained by proven impaired melatonin secretion in some patients with sleep disorders [22]. Indeed, the use of melatonin at low doses as a food supplement is allowed by the International Food Authorities to reduce the time it takes to fall asleep and to alleviate the subjective feeling of jet lag. However, although it is generally well tolerated [25], it is not always effective, and its possible adverse effects in the long term, especially in children and adolescents, are unknown [26,27]. 

In the last years, the consumption of supplements with melatonin has sharply increased, with some of them including concentrations of the hormone that exceed the ones declared on the label and reaching more than 30 mg/day [28]. This fact has produced an unexpected safety concern in territories with high melatonin consumption such as the USA, in which the frequent use of melatonin supplements by children after the COVID-19 pandemic has produced thousands of medical cases of unintentional poisoning, some of them with a fatal end [29]. For this reason, the use of other products that may increase melatonin levels during night-time without side effects, such as foods or nutritional supplements, could be an alternative to the use of this hormone to treat and/or prevent sleep disorders, especially in elderly people, in which melatonin secretion is markedly reduced [30].

Melatonin synthesis in the pinealocytes of the pineal gland starts from the conversion of tryptophan into serotonin by the enzyme tryptophan hydroxylase that transforms tryptophan into 5-hydroxytryptophan, which, in turn, is converted into serotonin. Serotonin is then acetylated by the enzyme arylalkylamine *N*-acetyltransferase (AANAT) to form *N*-acetylserotonin (NAS), which is the precursor of melatonin. This process is under the control of endocrine and neural systems which regulate the time, duration, and amount of melatonin produced [31]. Particularly, the increased activity of AANAT at the pineal gland during night-time is dependent on noradrenergic inputs sent from the suprachiasmatic nucleus of the hypothalamus by the sympathetic terminals of nervi conarii [32]. In pinealocytes, noradrenalin binds to beta and alpha adrenergic receptors and activates melatonin synthesis through the cAMP-PKA-CREB and PLC-Ca^2+^-PKC pathways [31]. 

Since melatonin is synthesized from tryptophan, tryptophan availability is crucial for melatonin synthesis. In humans, 95% of the tryptophan is metabolized in the liver through the kynurenine pathway (95%) [33]; therefore, only 5% of tryptophan is used for other purposes that include protein biosynthesis (around 4%) and the synthesis of serotonin and melatonin (around 1%) [34]. Tryptophan is converted into kynurenine by the enzymes indoleamine 2,3-dioxygenase (IDO) and tryptophan 2,3-dioxygenase (TDO). Subsequently, the metabolism of kynurenine can continue by several routes, which include the formation of kynurenine acid (KA) mediated by kynurenine aminotransferases (KAT), the formation of 3-hydroxykynurenine (3-HK) through kynurenine 3-monooxygenase (KMO), and the formation of 3-hydroxyanthranilic acid (3-HAA) through the enzyme kynureinase (KYNU). Anthranilic acid and 3-HK converge in the formation of quinolinic acid (QUIN), which is the precursor of NAD^+^ [35]. Thus, the main goal of the kynurenine pathway is the synthesis of NAD^+^, which is an important cofactor involved in many physiological processes. However, the hyperactivation of the kynurenine pathway may result in a decreased availability of tryptophan for serotonin and melatonin synthesis as well as in an increased production of metabolites, such as KA or QUIN, that are reported to exert neurotoxic effects [36]. In this regard, it is reported that, in sleep disorders, the increased levels of cortisol and proinflammatory cytokines induce the overexpression of the enzymes IDO and TDO, producing increased levels of neurotoxic QUIN through hyperactivation of the kynurenine pathway and reducing serotonin and melatonin levels [37]. 

A previous study in mice has reported that the effects of saffron as an antidepressant are the result of the reduced activation of the kynurenine pathway [38]. On the other hand, there some studies demonstrating the positive effects of saffron in improving sleep quality and duration in clinical trials [39,40,41,42]. Particularly, the beneficial effects of supplementation with low dosages of the same extract used in this study on sleep quality have been reported in three clinical trials performed by independent researchers in Japan [43] and Australia [39,44]. In the last study published in 2021 [39], the intake of 14 or 28 mg/day of affron^®^ one hour before bed increased melatonin levels at evening and improved sleep quality in healthy adults with unsatisfactory sleep. Although this study revealed a novel mechanism of action of saffron in sleep health, until now, no studies have been performed to determine if the increased melatonin levels after saffron supplementation are the result of increased melatonin synthesis and/or the decreased metabolism of tryptophan through the kynurenine pathway. Thus, the aim of this study was to analyze the effects of supplementation with the standardized extract of saffron (affron^®^) on the kynurenine pathway and melatonin synthesis in rats. 

## 2. Material and Methods

### 2.1. Materials

The saffron extract (affron^®^) was manufactured as previously described [39,44,45] by Pharmactive Biotech Products S.L.U. (Madrid, Spain) by a proprietary manufacturing process (patent ES2573542A1) and kindly donated to the researchers. Briefly, saffron was extracted with water at a temperature below 70 °C to preserve the activity of the sensitive active compounds safranal and crocins. The extract was lyophilized, grounded, and diluted with potato dextrin into a fine powder (sieved with a maximum mesh of 240 µm) to standardize it to 3.5% Lepticrosalides^®^, a term which refers to the sum of the said bioactive compounds of saffron, safranal, and crocin isomers, analyzed using HPLC [43]. Additionally, the extract contained picrorocin and kaempferol derivates, as was described in [43]. The extract was kept in darkness at room temperature until the experiment was performed.

### 2.2. In Vivo Study

#### 2.2.1. Animals

Sixteen 3-month-old male Wistar rats were fed ad libitum with a standard chow and housed under controlled temperature (22–24 °C) and humidity (50–60%) conditions. All the experiments were performed following the European Union Legislation and with the approval of the Animal Care Committee of the Universidad Autónoma de Madrid and the Autonomic Government of Madrid (PROEX 342.5-21). A strategy to limit to the minimum the number of animals used was applied (*n* = 8 rats/group).

#### 2.2.2. Treatment

Rats were treated once daily for seven days by oral gavage either with tap water (Control; *n* = 8) or with 150 mg/kg of affron^®^ dissolved in water one hour before the inactivity period. The volume of administration was 2.5 mL/kg in both experimental groups. The saffron extract (affron^®^) did not contain additives, and its soles contained the soluble saffron extract plus a soluble carrier in powder.

The rationale for dose selection was based on previous studies with botanic extracts that enhanced sleep in rats [46,47], particularly on a previous study in which the oral administration of 200 mg/kg of affron^®^ for 20 days exerted antianhedonic and mild antidepressant effects in male Wistar rats [45]. However, in this study, we wanted to assess if the beneficial effects of affron^®^ on sleep quality were present at a lower dose (150 mg/kg) and during a shorter period (7 days).

Body weight gain and food intake were checked daily as previously described [48]. On the day of sacrifice, animals were anesthetized with sodium pentobarbital (100 mg/kg) and killed by decapitation one hour after vehicle or affron^®^ administration. The trunk blood was collected, and the serum was obtained by centrifugation at 2000 rpm for 20 min. The pineal gland and the liver were immediately removed, weighed, and stored at −80 °C for further analysis.

### 2.3. Measurement of Subsantances in the Serum 

The lipid profile was determined by measuring the circulating levels of triglycerides, total cholesterol, low-density lipoprotein (LDL), and high-density lipoprotein (HDL) as previously described, using colorimetric assays from Spin React (Sant Esteve de Bas, Girona, Spain).

The serum levels of tryptophan, kynurenine (Immusmol, Bordeaux, France), melatonin, endothelin-1, noradrenaline (Elabscience, Houston, TX, USA), insulin (Merck Millipore, Darmstadt, Germany), interleukin-6 (Cusabio, Houston, TX, USA), and testosterone and corticosterone (Arbor Assays, Ann Arbor, MI, USA) were determined using ELISA kits according to the manufacturer’s instructions. 

### 2.4. RNA Extraction and Quantification

The tri-Reagent protocol was used to extract total RNA from the liver and the pineal gland [49]. The amount of total RNA was quantified using a Nanodrop 2000 (Thermo Fisher Scientifics, Hampton, NH, USA). Afterwards, cDNA was obtained from 1 µg of total RNA using a high-capacity cDNA reverse transcription kit (Applied Biosystems; Foster City, CA, USA).

### 2.5. Quantitative Real-Time PCR

Assay-on-demand Taqman probes (Applied Biosystems, Foster City, CA, USA) were used to measure the gene expression of arylalkyl-*N*-acetyltransferase (*Aanat*), aminoadipate aminotransferase (*Aadat*), indoleamine 2,3-dioxygenase 2 (*Ido-2*), tryptophan 2,3-dioxygenase 2 (*Tdo-2*), kynureinase (*Kynu*), interleukin 1β (*Il-1β*), tumoral necrosis factor α (*Tnf-α*) interleukin-6 (*Il-6*), heme oxygenase 1 (*Ho-1*), glutathione peroxidase (*Gpx-3*), glutathione reductase (*Gsr*), superoxide dismutase-1 (*Sod-1*), and the enzymes of the cytochrome P450 superfamily cytochrome, P450 1A2 (*Cyp1a2*) and 2C11 (Cyp2c11). The reference of each probe is shown in Table 1. Amplification was performed using TaqMan Fast Advanced Master Mix (Applied Biosystems, Foster City, CA, USA) following the manufacturer’s protocol in a StepOnePlus thermocycler (Applied Biosystems, Foster City, CA, USA). Relative gene expression was determined through the ΔΔCT method using the control group as a basal expression [50]. Results were normalized using the housekeeping gene hypoxanthine phosphoribosyltransferase (*Hprt-1*).

### 2.6. Statistical Analysis

Data are represented as mean ± SEM. Changes between variables in rats administered with vehicle or affron^®^ were analyzed using an unpaired Student’s *t*-test. Differences were considered significant when *p* < 0.05.

## 3. Results

### 3.1. Body Weight Gain and Food Intake

As shown in Figure 1, the treatment with the saffron extract significantly decreased body weight gain (*p* < 0.05) without producing significant changes in daily food intake.

### 3.2. Serum Measurements

Supplementation with affron^®^ for one week did not modify the circulating levels of triglycerides, total cholesterol, insulin, noradrenaline, interleukin-6, and corticosterone. However, treated rats showed decreased serum levels of LDL-c and ET-1 (Table 2; *p* < 0.05 for both) and increased serum concentrations of HDL-c and testosterone (Table 2; *p* < 0.05 for both).

### 3.3. Effects of affron^®^ Supplementation on the Circulating Levels of Tryptophan, Melatonin, and Kynurenine

As shown in Figure 2, supplementation with the saffron extract significantly increased the serum levels of melatonin and decreased the serum concentrations of kynurenine (*p* < 0.05 for both) without modifying the circulating levels of tryptophan. 

### 3.4. Gene Expression of Enzymes of the Kynurenine Pathway in the Pineal Gland

The gene expression of the enzymes of the kynurenine pathways *Aanat*, KATII (*Aadat*), *Ido-2*, and kynureinase (*Kynu*) in the pineal gland of rats supplemented either with vehicle or with affron^®^ is shown in Figure 3. 

Supplementation with the saffron extract did not modify the mRNA levels of *Aadat*, the enzyme that transforms kynurenine into kynurenic acid (Figure 3C). However, treated rats showed overexpression of *Aanat*, the limiting enzyme for the synthesis of melatonin in the pineal gland (Figure 3A; *p* < 0.05). Moreover, affron^®^ supplementation downregulated the gene expression of *Ido-2*, the enzyme that converts tryptophan into kynurenine (Figure 3B; *p* < 0.05), and significantly reduced the mRNA levels of *Kynu,* an enzyme that participates in the downstream conversion of kynurenine into 3-HAA, the precursor of QUIN (Figure 3D; *p* < 0.05).

### 3.5. Gene Expression of Proinflammatory Cytokines and Antioxidant Enzymes in the Pineal Gland

As shown in Figure 4, supplementation with the saffron extract did not modify the mRNA levels of *Il-1β* and *Tnf-α* in the pineal gland. However, the treated rats showed a downregulation in the gene expression of *Il-6* compared to rats administered with vehicle (*p* < 0.05) and an overexpression of the antioxidant enzymes *Ho-1*, *Sod-1*, *Gsr*, and *Gpx-3* (*p* < 0.05 for all) compared to rats administered with vehicle.

### 3.6. Gene Expression of Enzymes of the Kynurenine Pathway in the Liver

Figure 5 shows the mRNA levels of (A) indoleamine 2,3-dioxygenase 2 (*Ido-2*), (B) Tryptophan-2,3-dioxygenase 2 (*TDO-2*), (C) aminoadipate aminotransferase (*Aadat)*, and (D) kynureinase (*Kynu*) in the liver of rats administered with vehicle or affron^®^ for one week. The treatment did not modify the hepatic gene expression of *Kynu* but it significantly downregulated the gene expression of the two enzymes involved in the conversion of tryptophan into kynurenine *Ido-2* and *Tod-2* (*p* < 0.05 for both), as well as the gene expression of *Aadat*, which transforms kynurenine into KA.

### 3.7. Gene Expression of Proinflammatory Cytokines and Antioxidant Enzymes in the Liver

The hepatic mRNA levels of the proinflammatory cytokines interleukin-6 (*Il-6*), interleukin 1β (*Il-1β*), tumor necrosis factor alpha (*Tnf-α*) and the antioxidant enzymes hemooxygenase 1 (*Ho-1*), superoxide dismutase 1 (*Sod-1*), glutathione reductase (*Gsr*), and glutathione peroxidase 3 (*Gpx-3*) of rats administered with vehicle or with the saffron extract for one week is shown in Figure 6. 

Rats administered with affron^®^ did not show any differences in the hepatic gene expression of *Il-6* and *Sod-1* compared to rats administered with vehicle. However, the treatment significantly downregulated the gene expression of *Il-1β* and *Tnf-α* (*p* < 0.05 for both) and induced overexpression of *Ho-1*, *Gsr*, and *Gpx-3* (*p* < 0.05 for all) in the liver.

### 3.8. Gene Expression of Enzymes of the Cytochrome P450 Superfamily in the Liver

The hepatic gene expression of cytochrome P450 1A2 (*Cyp1a2*) involved in the metabolism of xenobiotics and cytochrome P450 2C11 (*Cyp2c11*) involved in the hepatic metabolism of steroid hormones and fatty acids are shown in Figure 7A and Figure 7B, respectively. 

The gene expression of *Cyp1a2* and *Cyp2c11* was similar in the liver of control and treated rats. 

## 4. Discussion

In this paper, we describe, for the first time, the possible mechanism of action of the standardized extract of saffron affron^®^ on mood and sleep quality. 

Our results show that supplementation with affron^®^ for one week to male rats one hour before the sleep period induced a significant increase in melatonin serum concentrations. These results agree with a previous study which demonstrated that supplementation with the same saffron extract for 28 days to adult men and women with unsatisfactory sleep one hour before bed results in a significant improvement in sleep quality that was associated with increased melatonin serum concentrations in the evening [39]. 

An important finding of the present study is that the circulating levels of melatonin after supplementation with the saffron extract were within the upper physiological range in rats but without producing hypermelatoninemia [51]. Thus, supplementation with affron^®^ may be useful to ameliorate the health concerns associated with hypomelatoninemia [52], particularly, the decreased melatonin serum concentrations associated with aging, but avoiding the possible adverse effects secondary to sustained increased melatonin levels. 

The effects of saffron supplementation on sleep quality may be due, at least in part, to the presence of crocin and its bioactive metabolite crocetin, since both carotenoids present in this saffron extract [53] are reported to increase the total time of non-rapid eye movement (non-REM sleep) [54], contributing to sleep maintenance and leading to improved subjective sleep quality [55]. Moreover, safranal may also contribute to affron^®^’s hypnotic effects, since safranal administration increases sleep duration in a dose-dependent manner [56].

To assess if the increased melatonin levels in the treated rats were the result of increased melatonin synthesis, we analyzed the gene expression of *Aanat* in the pineal gland and found significant overexpression of this enzyme in the rats supplemented with the saffron extract. These results are relevant because, although previous studies had reported increased melatonin concentrations after treatment with low dosages of affron^®^ (≥14 mg/day) in humans who self-reported poor sleep quality [39,43,44], until now, the mechanism of action was unexplored.

To investigate if the increased melatonin synthesis in the pineal gland was also associated with increased tryptophan availability due to a decreased conversion of tryptophan into kynurenine, the circulating levels of kynurenines, as well as the mRNA levels of the enzymes involved in the kynurenine pathway, were determined in the pineal gland and in the liver. Affron^®^ supplementation significantly reduced the gene expression of *Ido-2* both in the pineal gland and in the liver. Moreover, a significant downregulation of *Tdo-2* and *Aadat* was found in the liver, and a significant reduction in the mRNA levels of *Kynu* was also found in the pineal gland. These results demonstrate that the decreased kynurenine circulating levels are the result of the decreased activation of the kynurenine pathway in the liver, which is the organ that metabolizes 95% of tryptophan into kynurenine [33]. Moreover, the decreased activation of this pathway in the pineal gland may indicate a downregulation in the conversion of tryptophan into kynurenine which, together with the overexpression of *Aanat*, may result in an increased conversion of tryptophan into melatonin. These results are in agreement with a previous study which reported that the administration of a saffron extract to mice subjected to restrain stress exerted antidepressant effects through a decrease in the activation of the kynurenine pathway in different brain regions, which may result in a decreased production of certain neurotoxic metabolites from the kynurenine pathway [38]. In this regard, our results are relevant as they indicate that saffron administration not only improves the balance of melatonin/kynurenine in stressful situations but also in basal conditions. 

Since the hyperactivity of kynurenine pathway sleep disorders is secondary, at least in part, to increased levels of cortisol and proinflammatory cytokines [37], we analyzed if the affron^®^-induced decreased activation of this pathway was related to modifications in the circulating levels of interleukin-6 (IL-6) and corticosterone. However, our results show that supplementation with the saffron extract for one week did not modify either the serum levels of IL-6 or the circulating levels of corticosterone. These results agree with a previous study performed in healthy humans, in which increased melatonin levels, after affron^®^ administration for 28 days, were not associated with changes in the circulating levels of cortisol [39]. Likewise, in a study by Monchaux de Oliveira et al. in mice [38], the decreased activation of the kynurenine pathway after the administration of a saffron extract was not associated with decreased corticosterone levels. These results were, to some extent, expected, since saffron has been reported to modulate the activation of the hypothalamus–pituitary axis (HPA) under stressful situations but not in basal conditions. Indeed, saffron administration prevents an increase in corticosterone levels in mice subjected to foot shock stress but not in control animals [57]. 

Regarding IL-6, our results agree with those of a systematic review and meta-analysis which revealed that saffron supplementation does not have an impact on the circulating levels of IL-6 or in the plasma concentrations of other proinflammatory markers, such as the C reactive protein (CRP) or tumor necrosis alpha (TNF-α), in humans suffering from inflammatory processes due to different medical conditions [58]. However, treatment with a saffron extract significantly reduced the toxic effects and increased TNF-α circulating levels induced by diazinon in rats [59]. Moreover, the direct administration of crocin is reported to inhibit the production of IL-6 in vitro and in vivo in an experimental model of rheumatoid arthritis in mice [60]. Likewise, the direct administration of safranal for four weeks significantly reduced the circulating levels of interleukin 1β and TNF-α in an experimental model of type 2 diabetes in rats [61]. Thus, it is possible that the effects of saffron, in reducing the circulating levels of inflammatory markers, may depend on different factors, including the animal species, the inflammatory context, the duration of the treatment, and/or the concentrations of the bioactive compounds present in the saffron extract, such as crocin and safranal. 

Although affron^®^ did not reduce the serum levels of *Il-6*, we found a significant reduction in the gene expression of some proinflammatory cytokines both in the pineal gland and in the liver that may justify, at least in part, the decreased activation of the kynurenine pathway in both tissues. Particularly, in the pineal gland, affron^®^ supplementation downregulated the gene expression of *Il-6*, whereas, in the liver, the treated rats showed decreased mRNA levels of *Il-1β* and *TNF-α*. These results agree with other studies that have found decreased gene expression of proinflammatory markers after treatment with a saffron extract in several tissues, such as in the cerebral cortex [62], the hippocampus [63], the aorta [2,64], the retina [65], the kidney [66], the pancreas [67], and the liver [68,69]. However, to our knowledge, this is the first study to report an anti-inflammatory effect after saffron supplementation in the pineal gland. The effect of reducing the gene expression and/or protein levels of inflammatory markers could be due, at least in part, to the presence of crocin, a carotenoid that is reported to exert anti-inflammatory effects through the PI3K/AKT/mTOR pathway [70]. The affron^®^-induced downregulation of proinflammatory genes in the pineal gland may be related, at least in part, to its beneficial effect of improving mood [53] and sleep quality in humans [39,44], since, in both conditions, depression [71] and sleep disorders [72] are reported to occur in a proinflammatory context. However, whether the increased melatonin levels and increased melatonin synthesis found in treated rats are the results of decreased gene expression of proinflammatory cytokines in the pineal gland synthesis requires further investigation. 

In addition to inflammation, the other most characteristic effect of saffron is the antioxidant activity, which is due, at least in part, to the presence of crocetin, the active metabolite of crocin. This carotenoid is reported to inhibit ROS production by NOX-2 [73] and exert protective effects in several tissues through the Nrf2/HO-1/Keap1 signaling pathway [74].

For this reason, we analyzed the gene expression of the antioxidant enzymes *Ho-1*, *Gsr*, and *Gpx-3* both in the liver and in the pineal gland, finding that supplementation with the saffron extract significantly increased the antioxidant defenses in both tissues. The hepatic antioxidant effects of saffron have been widely reported in animal models and, again, seem to be due to the presence of crocin [75,76,77]. However, other bioactive compounds present in saffron such as safranal [78,79] and crocetin [80,81] are also reported to exert antioxidant properties in the liver, contributing to its hepatoprotective effects. In addition, a positive association has been found between the hepatic activation of the kynurenine pathway and liver failure [82]; therefore, the antioxidant effects, together with the decreased hepatic gene expression of the enzymes of the kynurenine pathway, may result in an improved liver function in affron^®^-treated rats. 

Regarding the antioxidant effects of affron^®^ in the pineal gland, this study is, again, the first one to report an upregulation of the antioxidant enzymes *Ho-1*, *Sod-1*, *Gsr*, and *Gpx-3*. Considering that both inflammation and oxidative stress are highly involved in the development of sleep disorders [83], the positive effects of affron^®^ on sleep quality are most likely due to its anti-inflammatory and antioxidant effects. However, whether reduced oxidative stress in the pineal gland results in increased melatonin synthesis needs to be further studied. Although the relationship between oxidative stress and melatonin synthesis is not completely understood, there is sufficient evidence demonstrating the link between increased ROS production and the development of neurological diseases, including depression. This association is due, at least in part, to the production of neurotoxic metabolites from the kynurenine pathway, which are reported to exert prooxidant effects [34]. Thus, the antioxidant effects of affron^®^ and the reduced production of metabolites from the kynurenine pathway are most likely related to its positive effects on mood and sleep disorders. Moreover, the improved melatonin concentrations may contribute to the health benefits due to its neuroprotector, anti-inflammatory, and antioxidant effects [84]. 

Since supplementation with saffron is reported to induce changes in the lipid profile and in the circulating levels of hormones involved in metabolism and cardiovascular and reproductive functions, the circulating levels of cholesterol, insulin, endothelin-1, and testosterone were analyzed. 

Supplementation with the saffron extract did not modify insulin serum concentrations, as previously described [85]. However, treated rats showed decreased circulating levels of LDL-c and increased serum concentrations of HDL-c. In addition to the improved lipid profile, affron^®^ administration significantly reduced the circulating levels of ET-1, a potent vasoconstrictor peptide produced by the vascular endothelium that favors the development of cardiovascular diseases such as hypertension. This result has already been reported both in experimental animals [86] and in hypertensive men [87] and may be related to the lower cardiovascular risk induced by saffron supplementation [88]. 

Finally, our results also show that supplementation with affron^®^ produces a strong increase in the circulating levels of testosterone. The positive effects of saffron on reproductive function have already been reported both in males [89] and in females [90] and are related to its anti-inflammatory and antioxidant effects in the reproductive system [89]. Likewise, saffron is also reported to exert beneficial effects on the urinary tract, which may also affect reproductive function [91]. In this regard, it has been reported that treatment with an extract obtained from *Serenoa repens*, *Crocus sativus*, and *Pinus massoniana* has a potent antioxidant and anti-inflammatory effect, preventing LPS-induced prostatitis ex vivo [92].

An important finding of our study is that supplementation with the saffron extract did not modify the gene expression of *Cyp1a2* or *Cyp2c11* in the liver, as had been previously described in rats [93] and in humans [94], thus avoiding the risk of interactions with co-administered substances metabolized by cytochrome P450 enzymes. 

In conclusion, supplementation with a standardized extract of saffron affron^®^ to male rats increases melatonin synthesis and decreases the mRNA levels of the enzymes of the kynurenine pathway in the pineal gland and in the liver, possibly through its anti-inflammatory and antioxidant effects. Moreover, affron^®^ administration increases the circulating levels of testosterone, improves the lipid profile, and decreases the circulating levels of ET-1 without modifying the hepatic gene expression of the enzymes of the cytochrome P450 superfamily. Considering that sleep disorders are associated with increased cardiovascular risk and with alterations in metabolic and reproductive functions, the positive effects of affron^®^ on melatonin levels may result in medium-term improved sleep quality and in improved reproductive and cardiometabolic health. 

## Figures and Tables

**Figure 1 antioxidants-12-01619-f001:**
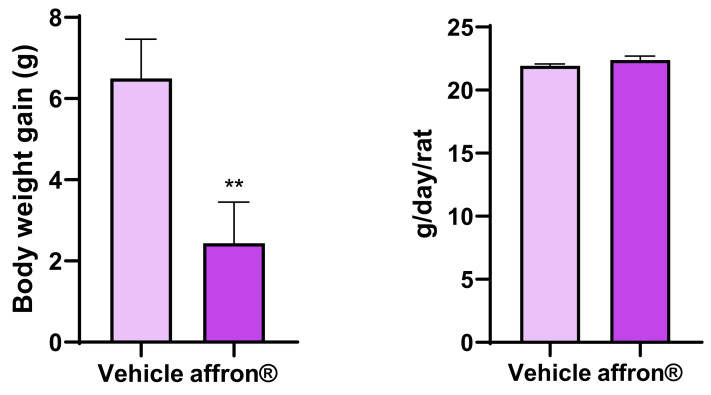
Body weight gain and daily food intake of rats administered with vehicle or affron^®^ for one week. Data are represented as the mean ± SEM. ** *p* < 0.01 vs. vehicle.

**Figure 2 antioxidants-12-01619-f002:**
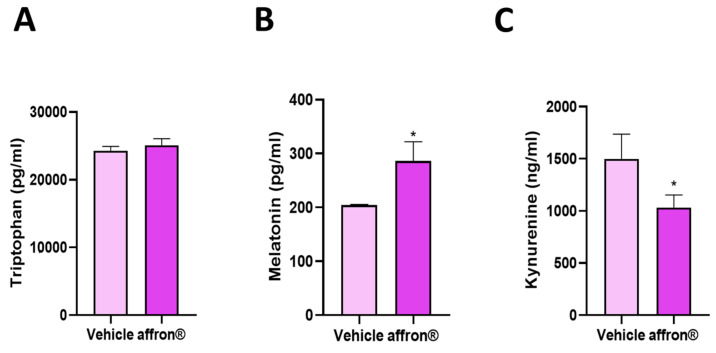
Serum levels of (**A**) tryptophan (pg/mL), (**B**) melatonin (pg/mL), and (**C**) kynurenine (ng/mL) in rats administered with vehicle or affron^®^ for one week. Data are represented as the mean ± SEM. * *p* < 0.05 vs. vehicle.

**Figure 3 antioxidants-12-01619-f003:**
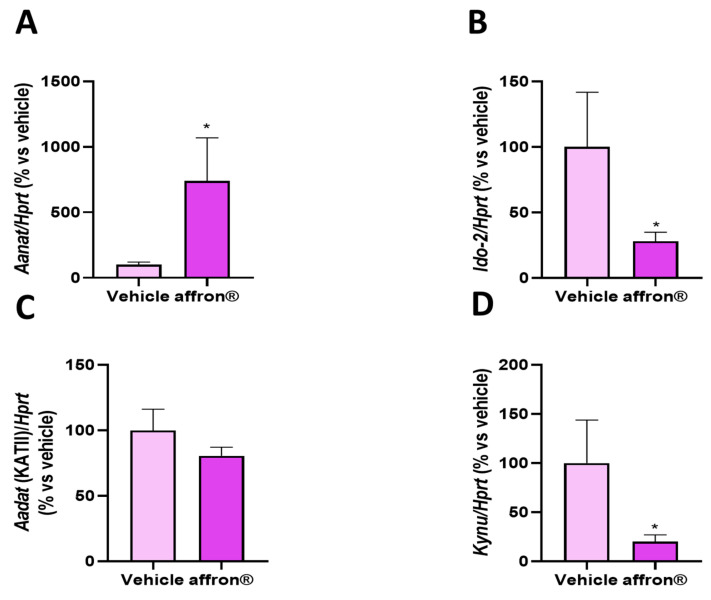
mRNA levels determined using qRT-PCR of arylalkyl-*N*-acetyltransferase (*Aanat*), aminoadipate aminotransferase *(Aadat)*, indoleamine 2,3-dioxygenase 2 (*Ido-2*), and kynureinase (*Kynu*) in the pineal gland of rats administered with vehicle or affron^®^ for one week. Data are represented as the mean ± SEM. * *p* < 0.05 vs. vehicle.

**Figure 4 antioxidants-12-01619-f004:**
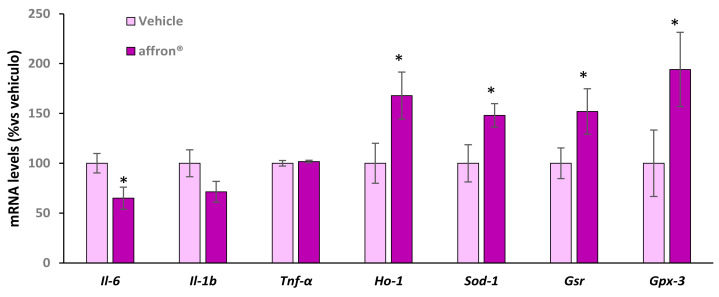
mRNA levels determined using qRT-PCR of interleukin-6 (*Il-6*), interleukin 1β (*Il-1β*), tumor necrosis factor Alpha (*Tnf-α*), hemooxygenase 1 (*Ho-1*), superoxide dismutase 1 (*Sod-1*), glutathione reductase (*Gsr*), and glutathione peroxidase 3 (*Gpx-3*) in the pineal gland of rats administered with vehicle or affron^®^ for one week. Data are represented as the mean ± SEM. * *p* < 0.05 vs. vehicle.

**Figure 5 antioxidants-12-01619-f005:**
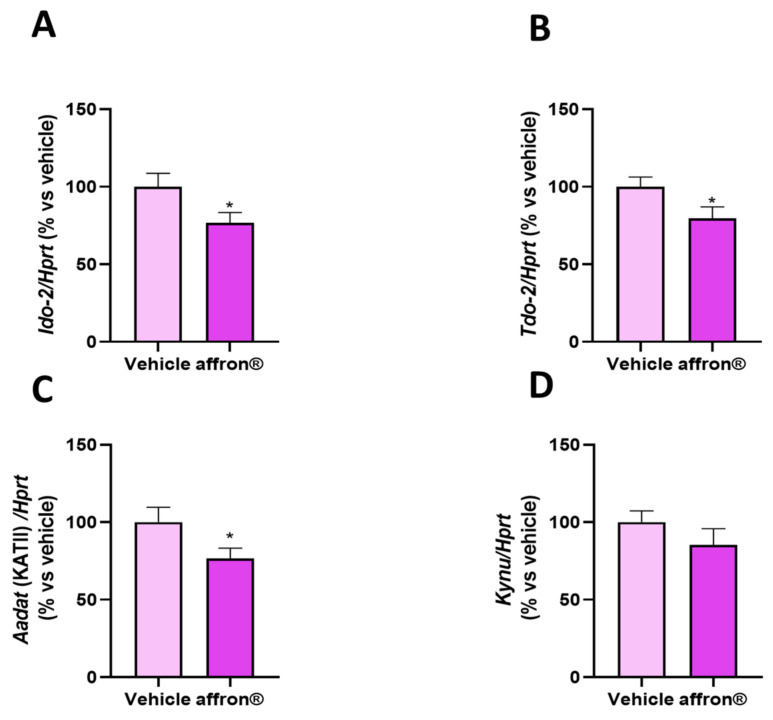
mRNA levels determined using qRT-PCR of (**A**) indoleamine 2,3-dioxygenase 2 (*Ido-2*), (**B**) Tryptophan-2,3-dioxygenase 2 (TDO-2), (**C**) aminoadipate aminotransferase *(Aadat)*, and (**D**) kynureinase (*Kynu*) in the liver of rats administered with vehicle or affron^®^ for one week. Data are represented as the mean ± SEM. * *p* < 0.05 vs. vehicle.

**Figure 6 antioxidants-12-01619-f006:**
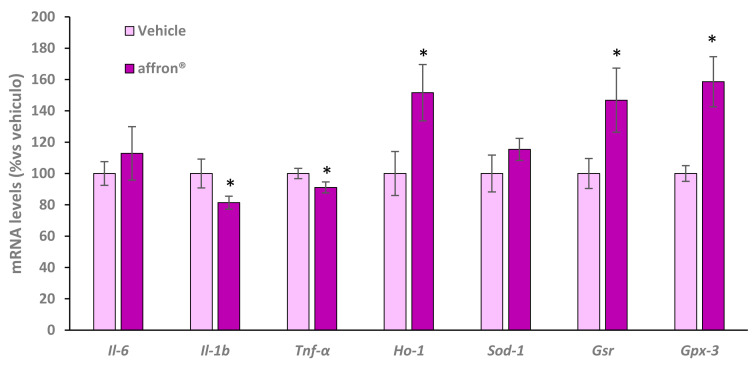
mRNA levels determined using qRT-PCR of interleukin-6 (Il-6), interleukin 1β (Il-1β), tumor necrosis factor alpha (Tnf-α), hemooxygenase 1 (Ho-1), superoxide dismutase 1 (Sod-1), glutathione reductase (Gsr), and glutathione peroxidase 3 (Gpx-3) in the liver of rats administered with vehicle or affron^®^ for one week. Data are represented as the mean ± SEM. * *p* < 0.05 vs. vehicle.

**Figure 7 antioxidants-12-01619-f007:**
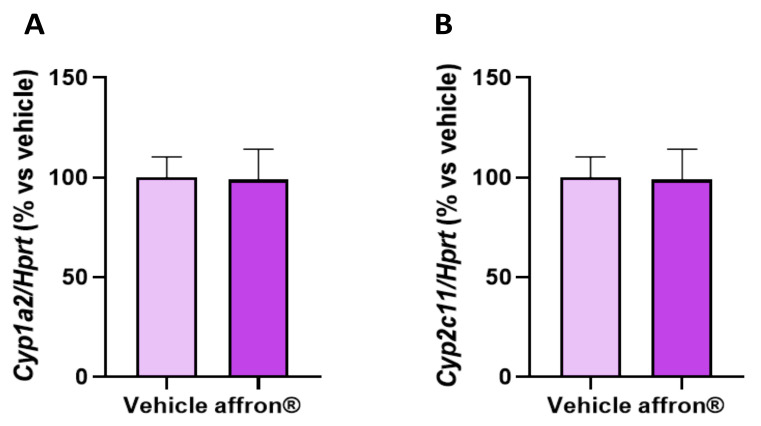
mRNA levels determined using qRT-PCR of (**A**) cytochrome P450 1A2 (*Cyp1a2*) and (**B**) cytochrome P450 2C11 (*Cyp2c11*) in the liver of rats administered with vehicle or affron^®^ for one week. Data are represented as the mean ± SEM.

**Table 1 antioxidants-12-01619-t001:** References of Taqman assays.

Genes	Reference
*Aadat*	Rn00567882_m1
*Aanat*	Rn00664873_g1
*Cyp1a2*	Rn00561082_m1
*Cyp2c11*	Rn01502203_m1
*Gpx-3*	Rn00574703_m1
*Gsr*	Rn01482159_m1
*Ho-1*	Rn00561387_m1
*Hprt-1*	Rn01527840_m1
*Il-1* *β*	Rn00580432_m1
*Il-6*	Rn01489669_m1
*Tnf-* *α*	Rn01525859_g1
*Kynu*	Rn01449532_m1
*Ido-2*	Rn01482543_m1
*Sod-1*	Rn00566938_m1
*Tod-2*	Rn00574499_m1

**Table 2 antioxidants-12-01619-t002:** Serum levels of triglycerides, total cholesterol, LDL–cholesterol (LDL-c), HDL–cholesterol (HDL-c), insulin, endothelin-1 (ET-1), noradrenaline (NA), interleukin-6 (IL-6), corticosterone, and testosterone in rats administered with vehicle or affron^®^. Data are represented as mean ± SEM. * *p* < 0.05 vs. vehicle.

	Vehicle	affron^®^
Triglycerides (mg/dL)	100 ± 4.3	121 ± 19.3
Total cholesterol (mg/dL)	87.7 ± 4.6	89.6± 5.8
LDL-c (mg/dL)	23.2 ± 1.8	19.2 ± 1.1 *
HDL-c (mg/dL)	25.3 ± 6.3	38.5 ± 3.3 *
Insulin (ng/mL)	2.7 ± 0.5	2.6 ± 0.5
ET-1 (pg/mL)	1.5 ± 0.1	1.3 ± 0.07 *
NA (ng/mL)	1.06 ± 0.1	0.92 ± 0.1
IL-6 (ng/mL)	3.2 ± 0.4	3.3 ± 0.2
Corticosterone (pg/mL)	258 ± 120	300 ± 102
Testosterone (ng/mL)	4.5 ± 2.1	15.2 ± 4.8 *

## Data Availability

Data are unavailable due to privacy restrictions.

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
