# Peer review of "Effects of Supplementation with the Standardized Extract of Saffron (affron®) on the Kynurenine Pathway and Melatonin Synthesis in Rats"

_antioxidants, 2023, doi:10.3390/antiox12081619_

Round 1
Reviewer 1 Report
L140 and 143: same extract used in this study (affron® ). I suggest to revise the sentence defining characteristic of the extract rather than the commercial name (standardized extract, characterized by…)
Par 2.1: Please add all relevant information on plant material and extraction preparation. If the manuscript is presented as focused on plant extracts, all relevant information for the reproducibility should be consider mandatory otherwise is only a testing on commercial product.
Par 2.2.1 please specify if the study was designed to limit to the minimum the number of animals
Par 2.2.2 please describe the treatment preparation. The commercial product form, aspect and composition should be described (solid or liquid?). Does the commercial products contain additives, technical ingredients or others?
Please specify the rationale for the dose selection.
Within the text the use of commercial product sounds sometimes redundant. Being only single treatment, I suggest to change in “treatment”
Uniform the style between text and figures affron/Affron
Uniform in the text/figures the untreated group as vehicle or control group
Please add references in the material and method sections for test and protocols used
I suggest to implement the general revision of saffron literature in the introduction with notes also on the new potential health use of non-stigma products from Crocus as well as innovative applications of the conventional stigmas. For example:
-doi: 10.1016/j.fct.2019.01.040
-DOI: 10.1016/j.foodres.2018.04.028
-Chiavaroli, A., Recinella, L., Ferrante, C., Locatelli, M., Carradori, S., Macchione, N., Zengin, G., Leporini, L., Leone, S., Martinotti, S., Brunetti, L., Vacca, M., Menghini, L., & Orlando, G. (2017). Crocus sativus, Serenoa repens and Pinus massoniana extracts modulate inflammatory response in isolated rat prostate challenged with LPS. Journal of biological regulators and homeostatic agents, 31(3), 531–541.
A revision on minor spelling and grammatical error is encouraged
Author Response
Reviewer 1
L140 and 143: same extract used in this study (affron® ). I suggest to revise the sentence defining characteristic of the extract rather than the commercial name (standardized extract, characterized by…)
As suggested by the reviewer, the commercial name of the extract has been deleted from this sentence. Its composition and method of extraction is now referenced in the Materials section (Lines 156-165)
Par 2.1: Please add all relevant information on plant material and extraction preparation. If the manuscript is presented as focused on plant extracts, all relevant information for the reproducibility should be consider mandatory otherwise is only a testing on commercial product.
In order to add the essential information required by the reviewer, the text has been modified as follows (Lines 156-165):
The saffron extract (affron®) was manufactured at the R&D facilities by Phar-mactive Biotech Products, SLU (Madrid, Spain). The hydrophilic saffron extract was prepared as previously described [37,42,43] by a proprietary manufacturing process. Briefly, saffron was extracted at controlled temperature in an aqueous solution to pre-serve the activity of sensitive active compounds. The extract was freeze dried carefully to a fine powder and diluted with a soluble carrier to standardize it to ≥3.5% Lep-ticrosalides®, a term which refers to the sum of the bioactive compounds of saffron, safranal and crocin isomers, analyzed by HPLC [43]. Additionally, the extract contains picrorocin, and kaempferol derivates, as it was described in [43]. The extract was kept in darkness at room temperature until the experiment was performed.
New reference (45):
Orio, L.; Alen, F.; Ballesta, A.; Martin, R.; Gómez de Heras, R. Antianhedonic and Antidepressant Effects of Affron®, a Standardized Saffron ( Crocus Sativus L.) Extract. Molecules. 2020, 25, 3207. DOI: 10.3390/molecules25143207.
Par 2.2.1 please specify if the study was designed to limit to the minimum the number of animals
Yes, the minimum number of animals was calculated prior to the experiment and established in 8 animals per group.
Par 2.2.2 please describe the treatment preparation. The commercial product form, aspect and composition should be described (solid or liquid?). Does the commercial products contain additives, technical ingredients or others?
As suggested by the reviewer the information regarding the form, aspect and composition of the commercial product is now stated in the following paragraph that has been included in the Materials and Methods section (Lines 174-177)
“Rats were treated once daily for seven days by oral gavage either with tap water (Control; n=8) or with 150 mg/kg of affron® dissolved in water one hour before the inactivity period. The volume of administration was 2,5ml/kg in both experimental groups. The saffron extract (affron®) does not contain additives, it soles contained the soluble saffron extract plus a soluble carrier in powder. “
Please specify the rationale for the dose selection.
As suggested by the reviewer the following paragraph regarding the rationale for dose selection has been included in the Materials and Methods section (Lines 179-184):
“The rationale for dose selection was based on previous studies with botanic extracts that enhanced sleep in rats (Kim et. al 2021, Jang et. al 2021) and particularly on a previous study in which the oral administration of 200 mg/kg of affron® for 20 days exerted antianhedonic and mild antidepressant effects in male Wistar rats (Orio et. al 2020) . However in this study we wanted to assess if the beneficial effects of affron® on sleep quality were present at a lower dose (150 mg/kg) and during a shorter period (7 days).
New references:
(45) Orio, L.; Alen, F.; Ballesta, A.; Martin, R.; Gómez de Heras, R. Antianhedonic and Antidepressant Effects of Affron®, a Standardized Saffron ( Crocus Sativus L.) Extract. Molecules. 2020, 25, 3207. DOI: 10.3390/molecules25143207
(46) Jang, H.S.; Jung, J.Y.; Jang, I.S.; Jang, K.H.; Kim, S.H.; Ha, J.H.; Suk, K.; Lee, M.G. L-theanine partially counteracts caffeine-induced sleep disturbances in rats. Pharmacol. Biochem. Behav., 2012, 101, 217-21. DOI: 10.1016/j.pbb.2012.01.011.
(47) Kim, S.; Hong, K.B.; Jo, K.; Suh, H.J. Quercetin-3- O-glucuronide in the Ethanol Extract of Lotus Leaf (Nelumbo nucifera) Enhances Sleep Quantity and Quality in a Rodent Model via a GABAergic Mechanism. Molecules, 2021, 26, 3023, DOI: 10.3390/molecules26103023.
Within the text the use of commercial product sounds sometimes redundant. Being only single treatment, I suggest to change in “treatment”
As suggested by the reviewer the name of the product (affron®) has been changed by “treatment” or by “saffron extract” in some paragraphs of the Results and Discussion sections.
Lines 229, 237, 246, 257, 258, 275, 276, 284, 296, 299, 312, 324, 328, 341, 343-344, 372, 402, 416, 425, 453, 454 and 470.
Uniform the style between text and figures affron/Affron
It has been corrected.
Uniform in the text/figures the untreated group as vehicle or control group
It has been corrected.
Please add references in the material and method sections for test and protocols used
In addition to the previous references about RNA extraction by the Tri-reagent protocol (49) and relative quantification of gene expression by the ΔΔCT method (50) the following reference about the procedure to measure food intake has been included in the Material and Methods section (Line 185)
New Reference:
- Gonzalez-Hedstrom, D.; Garcia-Villalon, A.L.; Amor, S.; de la Fuente-Fernandez, M.; Almodovar, P.; Prodanov, M.; Priego, T.; Martin, A.I.; Inarejos-Garcia, A.M.; Granado, M. Olive leaf extract supplementation improves the vascular and metabolic alterations associated with aging in Wistar rats. Sci Rep 2021, 11, 8188, doi:10.1038/s41598-021-87628-7.
I suggest to implement the general revision of saffron literature in the introduction with notes also on the new potential health use of non-stigma products from Crocus as well as innovative applications of the conventional stigmas. For example:
-doi: 10.1016/j.fct.2019.01.040
-DOI: 10.1016/j.foodres.2018.04.028
-Chiavaroli, A., Recinella, L., Ferrante, C., Locatelli, M., Carradori, S., Macchione, N., Zengin, G., Leporini, L., Leone, S., Martinotti, S., Brunetti, L., Vacca, M., Menghini, L., & Orlando, G. (2017). Crocus sativus, Serenoa repens and Pinus massoniana extracts modulate inflammatory response in isolated rat prostate challenged with LPS. Journal of biological regulators and homeostatic agents, 31(3), 531–541.
Following the instructions of the reviewer the following information and references have been added to the Introduction and Discussion sections:
Lines 49-51: “These compounds are present mainly in the stigmas although other parts of the plant such as the tepals and the anthers are also reported to be a good source of bioactive substances (Chichiricco et. Al 2019, Menghini et. Al 2018).”
Lines 465-469: “Likewise saffron is also reported to exert beneficial effects in the urinary tract which may also affect to the reproductive function (Quarto et. al 2017). In this regard it has been reported that treatment with an extract obtained from Serenoa repens, Crocus sativus and Pinus mas-soniana has a potent antioxidant and antiinflamatory effect preventing LPS-induced prostatitis ex vivo [Chiavaroli et. al 2017].”
New references:
- Chiavaroli, A.; Recinella, L.; Ferrante, C.; Locatelli, M.; Carradori, S.; Macchione, N.; Zengin, G.; Leporini, L.; Leone, S.; Martinotti, S., et al. Crocus sativus, Serenoa repens and Pinus massoniana extracts modulate inflammatory response in isolated rat prostate challenged with LPS. J Biol Regul Homeost Agents 2017, 31, 531-541.
- Chichiricco, G.; Ferrante, C.; Menghini, L.; Recinella, L.; Leone, S.; Chiavaroli, A.; Brunetti, L.; Di Simone, S.; Ronci, M.; Piccone, P., et al. Crocus sativus by-products as sources of bioactive extracts: Pharmacological and toxicological focus on anthers. Food Chem Toxicol 2019, 126, 7-14, doi:10.1016/j.fct.2019.01.040.
- Menghini, L.; Leporini, L.; Vecchiotti, G.; Locatelli, M.; Carradori, S.; Ferrante, C.; Zengin, G.; Recinella, L.; Chiavaroli, A.; Leone, S., et al. Crocus sativus L. stigmas and byproducts: Qualitative fingerprint, antioxidant potentials and enzyme inhibitory activities. Food Res Int 2018, 109, 91-98, doi:10.1016/j.foodres.2018.04.028.
- Quarto, G.; Cola, A.; Perdona, S. Efficacy of a formulation containing Serenoa repens, Crocus sativus and Pinus massoniana extracts in men with concomitant LUTS and erectile dysfunction. Minerva Urol Nefrol 2017, 69, 300-306, doi:10.23736/S0393-2249.16.02661-8.

Reviewer 2 Report
Dear Editor,
Please, find below a revision of the manuscript entitled: “Effects of supplementation with the standardized extract of saffron (affron®) on the kynurenine pathway and melatonin synthesis in rats” submitted to the journal - Antioxidants.
Sleep disorders affect approximately 20 percent of adults and are associated with alterations in cognitive functions, stress responsivity, memory, and emotional reactivity. Due to its high prevalence, the use of drugs for insomnia treatment has risen in recent years. On the other hand, increased social awareness causes new therapeutics questing, particularly among natural compounds origin. Therefore, this work conforms to new research trends.
Generally, the work is interesting; however, some sentences are unclear or seem unfinished. Thus, minor English language editing is required, as well as some corrections in the methodology and results.
Line 117 – This is the first sentence of the phrase, and the author shows: "Since melatonin is ... for this process" – what process? It should be explained.
Section 2.3. title - it is not Serum measurements but rather serum metabolites determination - it should be specified.
The results section is the weakest part of the manuscript. The author did not describe results but only showed (e.g., line 218): affron-treated rats showed decreased serum levels of LDL-c and ET-1.
The author should instead show: what high were changes?
In Figures 3 - 5 titles, the author shows mRNA level. Is it true? Was the mRNA level examined by Northern blot or after agarose gel electrophoresis? If not, thus, the author determined the relative expression level by qRT-PCR (line 199). It should be corrected in the Figures and the text of the manuscript.
Minor editing of the English language is required because some sentences are unclear or seem to be not finished.
Author Response
Line 117 – This is the first sentence of the phrase, and the author shows: "Since melatonin is ... for this process" – what process? It should be explained.
The sentence “Since melatonin is synthetized from tryptophan, tryptophan availability is crucial for this process” has been changed by “Since melatonin is synthetized from tryptophan, tryptophan availability is crucial for melatonin synthesis.” (Line 119)
Section 2.3. title - it is not Serum measurements but rather serum metabolites determination - it should be specified.
“Serum measurements” has been changed by “Measurement of substances in the serum”
The results section is the weakest part of the manuscript. The author did not describe results but only showed (e.g., line 218): affron-treated rats showed decreased serum levels of LDL-c and ET-1. The author should instead show: what high were changes?
We are sorry but we do not understand well what the reviewer means by “not describe results but only showed (e.g., line 218)” and by “what high were changes?”
In the results section we describe the effects of affron® supplementation compared to the administration of vehicle. Particularly in line 218 we mean that the circulating levels of both ET-1 and LDL-c are significantly lower in rats treated with affron® compared to rats administered with vehicle.
In Figures 3 - 5 titles, the author shows mRNA level. Is it true? Was the mRNA level examined by Northern blot or after agarose gel electrophoresis? If not, thus, the author determined the relative expression level by qRT-PCR (line 199). It should be corrected in the Figures and the text of the manuscript.
The mRNA levels were determined by qRT-PCR as it is stated and defined in the Material and Methods section. To make it clearer we have now stated this in each Figure description.

Round 2
Reviewer 1 Report
I appreciate the revision of the manuscript but still some perplexity survive to the trend of proposing the manuscript as based on plant derived extracts rather than define it as a commercial product.
The key difference is that if authors investigate an herbal extract all information to permit to a different researcher to do similar work is (should be) mandatory. When a commercial standardized product is used, the commercial name and brand give the guarantee of similar qualitative standard (covered probably by the process patent).
Do authors are able to produce similar extract from plant material ?
What is intended for:
- hydrophilic extraction?
-controlled temperature ?
- was freeze dried carefully?
-fine powder?
-soluble carrier?
-guarantee the unaltered presence of “sensitive” compounds?
I can access only to one of the new references (37, 42, 43) and is reported” Affron® is a patented compound (ES2573542A1)” and not details of the preparation.
Furthermore, as authors state, the effects is related to the concentration of Affron® that do not directly reflect the concentration of extracts (Affron® powder is extract+carrier).
I suggest to add in material and methods that a strategy to limit to the minimum the number of animal used was applied.
Info on the origin of tested sample is missing (do authors buy (where)? gift from company?
Author Response
I appreciate the revision of the manuscript but still some perplexity survive to the trend of proposing the manuscript as based on plant derived extracts rather than define it as a commercial product. The key difference is that if authors investigate an herbal extract all information to permit to a different researcher to do similar work is (should be) mandatory. When a commercial standardized product is used, the commercial name and brand give the guarantee of similar qualitative standard (covered probably by the process patent).
The authors confirm that affron® is a commercial saffron extract made by a concentrated saffron extract and a soluble carrier to yield a final concentration of active ingredients of at least 3.5% Lepticrosalides®. The product is used as a functional ingredient in the manufacturing of food supplements or pharma products.
Do authors are able to produce similar extract from plant material ?
The authors can access to the identical commercial product manufactured by Pharmactive Biotech Products, SLU without restriction.
The basic protocol with the description of the critical extraction parameters is described in the patent ES2573542, owned by Pharmactive Biotech Products. This information guarantees the production of saffron extracts similar to the one used in this study.
A brief description of the production process is now stated in the materials section (Lines 156-165)
“The saffron extract (affron®) was manufactured as previously described [39,44,45] by Pharmactive Biotech Products SLU (Madrid, Spain) by a proprietary manufacturing process (patent ES2573542A1) and kindly donated to the researchers. Briefly, saffron was extracted with water at temperature below 70°C to preserve the activity of the sensitive active compounds safranal and crocins. The extract was lyophilized, grinded and diluted with potato dextrin to a fine powder (sieved with a maximum mesh of 240 µm) to standardize it to 3.5% Lepticrosalides®, a term which refers to the sum of the said bioactive compounds of saffron, safranal and crocin iso-mers, analyzed by HPLC [43]. Additionally, the extract contains picrorocin, and kaempferol derivates, as it was described in [43]. The extract was kept in darkness at room temperature until the ex-periment was performed.”
What is intended for:
- hydrophilic extraction?
The extraction process is described in the patent ES2573542 owned by Pharmactive Biotech Products. As stated in the patent, the extraction process is carried out with “water or hydroalcoholic mixtures” in which the proportion of ethanol does not exceed 20 % (v/v). The extraction is performed using polar solvents with the aim of extracting preferably water soluble compounds, particularly crocins, which are water soluble carotenoids.
-controlled temperature ?
As is described in the mentioned patent, “temperature control during the extraction process is critical and it must be limited to 70°C in order to avoid the deterioration of certain crocin isomers that could be affected under higher temperatures” (literally from the patent).
Consequently, the extraction was done at temperature ≤70°C.
- was freeze dried carefully?
The lyophilization temperature did not surpass 20°C
-fine powder?
As is described in the mentioned patent, the commercial product was sieved “through a sieve with a maximum mesh of 240 µm”. Hence, the product was sieved to yield field powder with a maximum mesh of 240 μm.
-soluble carrier?
The patent claims the use of “pharmaceutically acceptable excipients until the formula of the present invention is obtained. Non-limiting examples of excipients are starch, lactose, dextrose, maltodextrin, sucrose, mannitol, sorbitol, glucose, microcrystalline cellulose, di- and tricalcium phosphate, calcium sulfate, Kaolin and sodium chloride. Preferably, the excipient is obtained from dextrin, maltodextrin or any of their mixtures”.
-guarantee the unaltered presence of “sensitive” compounds?
The active compounds are safranal and crocins grouped to the trademark Lepticrosalides®, as described in the chapter 2.1.
As stated above, crocins are temperature sensitive compounds. Safranal (2,2,6-trimethyl-1,3-cyclohexadiene-1-carboxaldehyde) accounts for more than 60% of the total aromatic compounds and is the main component responsible for the flavor and aroma of saffron. As it is stated in the patent, safranal is a volatile compound which vanishes in the air at relatively low temperature. Hence, the manufacturing process is carefully designed to prevent the deterioration of crocins or the evaporation of safranal.
I can access only to one of the new references (37, 42, 43) and is reported” Affron® is a patented compound (ES2573542A1)” and not details of the preparation.
The patent information is available in the following link:
https://patents.google.com/patent/ES2573542A1/en
However, a brief description of the production process is now stated in the materials section (Lines 156-165)
“The saffron extract (affron®) was manufactured as previously described [39,44,45] by Pharmactive Biotech Products SLU (Madrid, Spain) by a proprietary manufacturing process (patent ES2573542A1) and kindly donated to the researchers. Briefly, saffron was extracted with water at temperature below 70°C to preserve the activity of the sensitive active compounds safranal and crocins. The extract was lyophilized, grinded and diluted with potato dextrin to a fine powder (sieved with a maximum mesh of 240 µm) to standardize it to 3.5% Lepticrosalides®, a term which refers to the sum of the said bioactive compounds of saffron, safranal and crocin iso-mers, analyzed by HPLC [43]. Additionally, the extract contains picrorocin, and kaempferol derivates, as it was described in [43]. The extract was kept in darkness at room temperature until the ex-periment was performed.”
Furthermore, as authors state, the effects is related to the concentration of Affron® that do not directly reflect the concentration of extracts (Affron® powder is extract+carrier).
The authors wish to remind that the commercial product affron® is a standardized saffron extract. Standardization pursues to avoid the variability in active compounds concentration batch-to-batch, which is typical of plant extraction. It guarantees a minimum value of % Lepticrosalides® (safranal + crocins) by the addition of enough quantities of an inert soluble carrier to a concentrated saffron extract. Therefore, affron® is a saffron extract with a known concentration in active compounds. Since the content of Lepticrosalides® is known and guaranteed, the effects of the product are related to the concentration of the active compounds.
For the sake of clarity, the concentration of active compounds in the batch of affron® used in the assays was 3,5% Lepticrosalides® as stated in the Materials section (Line 162).
I suggest to add in material and methods that a strategy to limit to the minimum the number of animal used was applied.
As suggested by the reviewer the following sentence has been added to the material and methods section (Lines 172-173)
“A strategy to limit to the minimum the number of animal used was applied (n=8 rats/group)”
Info on the origin of tested sample is missing (do authors buy (where)? gift from company?
As stated in the materials section (Lines 156-158) “The saffron extract (affron®) was manufactured as previously described [39,44,45] by Pharmactive Biotech Products SLU (Madrid, Spain) by a proprietary manufacturing process (patent ES2573542A1) and kindly donated to the researchers.”
